# The association between dietary inflammatory index and non-alcoholic fatty liver disease: A systematic review and meta-analysis

**Jingjing Lin** *, **Mengna Huang, Lina Shen**

Department of Infectious Diseases and Hepatology, Ningbo Medical Center Lihuili Hospital, Ningbo, Zhejiang, China

* 195554409@qq.com

## Abstract

### Background

The Dietary Inflammatory Index (DII) is a literature-based tool designed to predict inflammation. Previous studies suggest a potential association between the DII and non-alcoholic fatty liver disease (NAFLD). However, the relationship between the DII and both the incidence and progression of NAFLD remains unclear.

### Methods

Systematic literature searches were conducted in PubMed, Web of Science, Embase, Scopus, and the Cochrane Library up to July 2025. A random-effects model was applied, and odds ratios (ORs) with 95% confidence intervals (CIs) were calculated. Sensitivity and subgroup analyses were performed to explore the sources of heterogeneity, while Egger's test was used to assess publication bias. Review Manager 5.4 and STATA 15.0 were employed for statistical analysis.

### Results

Eighteen studies involving 262,468 participants were included. The data indicated a significant association between the DII and NAFLD (OR = 1.33, 95% CI: 1.23–1.44; $P < 0.00001$) and between the DII and fibrosis (OR = 1.36, 95% CI: 1.20–1.54; $P < 0.00001$). Subgroup analysis identified geographic region and diagnostic criteria as sources of heterogeneity. Egger's test revealed publication bias for NAFLD.

### Conclusion

A high DII was associated with an increased risk of NAFLD and an increased risk of progression to fibrosis.

**Data availability statement:** All relevant data are within the paper and its Supporting Information files.

**Funding:** The author(s) received no specific funding for this work.

**Competing interests:** The authors have declared that no competing interests exist.

## Systematic review registration

PROSPERO, identifier CRD42025632168.

## 1. Introduction

Non-alcoholic fatty liver disease (NAFLD) is a common chronic liver disorder that ranges from benign fat accumulation to non-alcoholic steatohepatitis (NASH) and may progress to liver fibrosis or cirrhosis. [1]. The prevalence of NAFLD is 20–30% in Western populations, 5–18% in Asian populations [2], and 60–70% in patients with type 2 diabetes [3,4]. The growing rates of NAFLD are concerning due to its association with a higher risk of mortality from end-stage liver disease and other related complications [5,6]. Given the substantial health implications, NAFLD screening plays a key role in preventing its development into advanced liver fibrosis. Although liver biopsy is the standard diagnostic method, its impracticality in large populations has led to the widespread adoption of non-invasive techniques [7]. NAFLD is influenced by metabolic, genetic, environmental, and gut microbiome factors [8], with diet serving as a key modifiable factor in liver health. The Dietary Inflammatory Index (DII) summarizes the effects of diet on inflammatory markers [9], drawing form hundreds of studies linking diet to biomarkers such as IL-1β, IL-4, IL-6, IL-10, TNF-α, and C-reactive protein (CRP) [10–13]. The DII assesses 45 parameters, including 35 nutrients, to evaluate a diet's inflammatory potential, with higher scores indicating a more pro-inflammatory diet [14]. Previous studies have linked dietary inflammatory potential to NAFLD [15–16], and the DII not only predicts NAFLD but also the risk of other chronic conditions, including cancer [17], diabetes [18], depression [19], cardiovascular disease, and both overall and cause-specific mortality [20]. Wu et al. [21] identified the DII as a risk factor for increased hypertension prevalence. Compared to tools like the DIS, the DII offers a broader scientific basis, a well-established application in cancer research, a comprehensive evaluation of ingredients, and a strong correlation with inflammatory markers, making it the preferred tool for studying the relationship between dietary inflammation and cancer. Amirkalali B et al. [22] published a systematic review and meta-analysis on the dietary inflammatory index and the risk of non-alcoholic fatty liver disease, which concluded that an anti-inflammatory diet can reduce the risk of NAFLD. However, the relationship between the DII and progression of NAFLD remains unclear. This study aims to further explore the association between the DII and NAFLD and fibrosis by incorporating new evidence, expanding the study's scope, and uncovering previously overlooked clinical heterogeneity through subgroup analysis. Although recent guidelines (2023 Delphi Consensus) from major hepatology societies (AASLD, EASL, and ALEH) recommend transitioning to the term metabolic dysfunction-associated steatotic liver disease (MASLD) [23,24], because MASLD incorporates broader metabolic dysfunction criteria (e.g., cardiovascular risk factors) and reflects an improved understanding of metabolic contributors to fatty liver disease. However, since our analysis is based on historical studies using NAFLD criteria, this study retains the traditional NAFLD nomenclature to maintain consistency with the existing literature.

## 2. Methods

The study adhered to PRISMA guidelines [25] and was registered with PROSPERO (CRD42025632168). The registration occurred on December 28, 2024, and a literature search was initiated at the time of registration; however, literature screening had not yet been performed at that stage (S1 Table).

### 2.1 Literature search strategy

Studies on the relationship between the DII and NAFLD were searched in PubMed, Web of Science, Embase, Scopus, and the Cochrane Library up to July 2025. The database search included the following terms: (Nonalcoholic Fatty Liver Disease OR NAFLD OR Fatty Liver, Nonalcoholic OR Fatty Livers, Nonalcoholic OR Liver, Nonalcoholic Fatty OR Livers, Nonalcoholic Fatty OR Nonalcoholic Fatty Liver OR Nonalcoholic Fatty Livers OR Nonalcoholic Fatty Liver Disease OR Nonalcoholic Steatohepatitis OR Nonalcoholic Steatohepatitides OR Steatohepatitides, Nonalcoholic OR Steatohepatitis, Nonalcoholic OR fatty liver OR MASLD OR metabolic dysfunction-associated steatohepatitis OR MASH) AND (dietary inflammatory index OR dietary inflammatory score OR DII OR dietary inflammatory). Additionally, two researchers reviewed the reference lists to ensure comprehensive coverage. The detailed retrieval strategy is presented in the S2 Table.

### 2.2 Inclusion and exclusion criteria

Two researchers independently evaluated articles for inclusion based on the following criteria:
P: Patients diagnosed with NAFLD;
E: Intervention group with higher DII;
C: Control group with lower DII;
O: Studies reporting DII and NAFLD outcomes;
S: Observational studies or randomized controlled trials (RCTs).
Studies were excluded based on the following criteria:

(i)   Full texts unavailable;

(ii)  Data not available;

(iii) For studies involving the same cohort, the most comprehensive or most recent article was included.

In case of disagreement, the researchers discussed the issue and resolved it, or consulted a third reviewer to assist in reaching a decision.

### 2.3 Data extraction

The extracted data included the following: first author's name, publication year, study period, region, study design, demographics (gender, age), diagnostic criteria, BMI, DII measurement method, DII cut-off, and primary outcomes (NAFLD). Secondary outcome measures included fibrosis, frailty, and ORs with 95% CIs for prognostic assessment. Data extraction was performed by MN Huang and LN Shen, with disagreements resolved through discussion. Unresolved issues were addressed by the senior author J.J. Lin.

### 2.4 Quality assessment

The Newcastle-Ottawa Scale (NOS) was used to assess the quality of the included cohort and case-control studies [26], with studies scoring 7–9 points considered high quality. Quality assessment of all included studies was independently conducted by two authors, and discrepancies were resolved through discussion.

## 2.5 Statistical analysis

The analysis employed odds ratios (ORs) with 95% confidence intervals (CIs) to examine the association between the Dietary Inflammatory Index (DII) and non-alcoholic fatty liver disease (NAFLD). Heterogeneity among studies was evaluated using the chi-square test, with $I^2$ values ≤50% considered indicative of low heterogeneity and values ≥50% representing substantial heterogeneity [27]. Due to the potential heterogeneity among the included studies, a random-effects model was applied to enhance the credibility of the results. Publication bias was evaluated using funnel plots and Egger's test ($P < 0.05$) [28]. Additionally, sensitivity analyses were conducted to evaluate the stability of the results, and subgroup analyses were used to explore the sources of heterogeneity. A two-tailed P-value $< 0.05$ was considered statistically significant. STATA 15.0 and Review Manager 5.4 were used for statistical analysis.

## 3. Results

### 3.1 Study selection and data extraction

Fig 1 presents an overview of the literature selection process. Titles, abstracts, and full manuscripts were screened to identify relevant studies. Initially, 8781 records were identified, and after removing duplicates, 7574 records remained. Titles and abstracts were then reviewed, resulting in the exclusion of 7546 studies. Of the remaining 28 articles, 6 reviews, 2 letters to the editor, 1 animal study, and 1 article with unextractable data were excluded. Ultimately, 18 articles were included. Based on the DII cut-off, 43 comparison groups were extracted from these 18 studies.

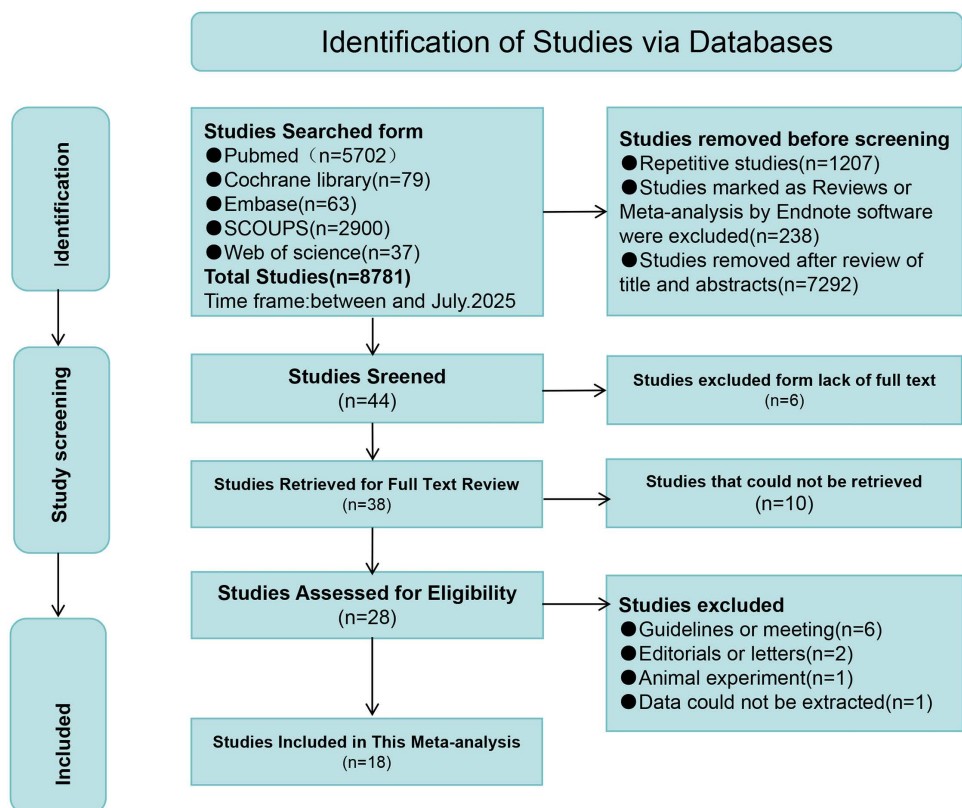

**Fig 1. Flowchart of literature screening.**

## 3.2 Study characteristics

A total of 262,468 patients from 18 studies [7,29–45] were included. Basic information is provided in Table 1. The included observational studies were published from 2018 to 2024. Two were cohort studies [37,38], and the remainder were case-control studies [7,29–36,39–45]. Six studies [7,36–40] focused on patients aged 50 or older, while twelve included patients aged 50 or younger [29–35,41–45]. The outcomes varied: sixteen studies [7,29–33,35,37–45] assessed NAFLD incidence, four [34,35,40,41] examined fibrosis, and one study [36] assessed frailty. The Newcastle-Ottawa Scale [26] indicated consistent study quality, with each study scoring above 6, suggesting high quality and a low risk of bias. S3 Table summarizes the study characteristics and quality.

## 3.3 Meta-analysis results

### 3.3.1 DII and NAFLD. 
We assessed the relationship between the DII and NAFLD by analyzing 37 comparison groups with 241,309 participants. Significant heterogeneity was observed ($I^2 = 74\%$, $P < 0.0001$), and a random-effects model was employed (Fig 2). The results showed a significant association between the DII and NAFLD (OR = 1.33, 95% CI: 1.23–1.44; $P < 0.0001$) (Fig 2). A higher DII score indicates a more proinflammatory diet, which has been linked to a higher likelihood of developing NAFLD. To address confounding factors and explain the heterogeneity, subgroup analyses were conducted based on average age (>50 or ≤50 years), study design (case-control or cohort), sample size (>5000 or ≤5000), geographic region (Asia, Europe, or the Americas), and diagnostic criteria for NAFLD. Table 2 summarizes these findings. Heterogeneity in NAFLD was mainly attributed to geographic region (OR = 1.11, 95% CI: 1.03–1.20; $P = 0.004$, $I^2 = 10\%$) and diagnostic criteria for NAFLD (OR = 1.37, 95% CI: 1.19–1.57; $P < 0.00001$, $I^2 = 49\%$).

### 3.3.2 DII and fibrosis. 
Twelve comparison groups provided data on the relationship between the DII and fibrosis, which were analyzed using a random-effect model (Fig 3). A significant positive correlation was found between the DII and fibrosis (OR = 1.36, 95% CI: 1.20–1.54; $P < 0.0001$, Fig 3). Elevated DII scores have been linked to a greater risk of progression to advanced fibrosis. To address confounding factors and explore heterogeneity, subgroup analyses were conducted, stratifying studies by average age (>50 or ≤50 years), study design (case-control or cohort), sample size (>5000 or ≤5000), geographic region (Asia, Europe, or the Americas) and diagnostic criteria for fibrosis. Table 2 summarizes the results. The findings indicated that fibrosis is linked with sample size (OR = 1.40, 95% CI: 1.23–1.60; $P < 0.00001$), suggesting that larger sample sizes (>5000) yield more stable results, furthermore, the fatty liver index was found to be more accurate in detecting the degree of liver fibrosis in clinical practice (OR = 1.50, 95% CI: 1.29–1.75; $P < 0.00001$).

### 3.3.3 DII and frailty. 
The study [36] suggests that frailty is characterized by a decrease in physiological reserve and an increased susceptibility to adverse health outcomes. In the elderly, it is commonly linked to functional impairments, a rise in disability, and an increased risk of mortality. Two comparison groups examined the association between the DII and frailty, which were analyzed using a random-effects model (Fig 4). A significant positive correlation was found between the DII and frailty (OR = 1.21, 95% CI: 1.09–1.35; $P = 0.0004$, Fig 4). However, due to the limited sample size, the results are considered exploratory, and more research is needed in the future to verify them.

## 3.4 Sensitivity analysis and publication bias

The stability of the DII results was assessed through sensitivity analyses. Removing each study sequentially showed consistent effect sizes, indicating that no individual study significantly influenced the results for NAFLD (S1Fig) and fibrosis (S2Fig), thus confirming the reliability of the findings. To evaluate publication bias in the relationship between DII and both NAFLD severity and liver fibrosis, funnel plots and Egger's tests were employed. Publication bias was detected for NAFLD (Egger: p = 0.014, Fig 5), but not for fibrosis severity (Egger: p = 0.054, S3Fig). The effect of publication bias on NAFLD was further assessed using the trim-and-fill method. The results showed that the correlation between the DII and NAFLD remained significant after trimming (OR = 1.205, 95% CI: 1.107–1.311) (Fig 6), indicating that the results of DII and NAFLD remain stable, suggesting a possible association between the two.

**Table 1. Basic characteristics of included literature.**

| Author | Study period | Country | Population | Study design | Diagnostic criteria | DII measurement method | Male/ Female | Mean age (yrs) | Mean BMI | DII cut-off | Patients NO | Outcome Measure | Quanlity score |
|---|---|---|---|---|---|---|---|---|---|---|---|---|---|
| Vahid2018a | 2015 | Iran | NAFLD patients and controls | case-control study | Ultrasound | 24-hour recalls, Food Frequency Questionnaire (FFQ), the7-day dietary recall (7DDR) and food records/diet diaries | NA | 43 | 29.75 | −1.94 | 999 | NAFLD | 8 |
| Vahid2018b | 2015 | Iran | NAFLD patients and controls | case-control study | Ultrasound | | NA | 43 | 29.75 | −0.87 | 999 | NAFLD | 8 |
| Mazidi2018 | 2006 | USA | NAFLD patients and controls | case-control study | Fatty liver index | 24-h dietary recalls | 10053/10590 | 48 | 28 | NA | 20643 | NAFLD | 7 |
| Ramírez-Vélez2021a (Tertile1) | 2017-2018 | USA | NAFLD patients and controls | case-control study | Transient elastography | 24-h dietary recalls | 2057/2132 | 67 | 30 | NA | 4189 | NAFLD | 7 |
| Ramírez-Vélez2021b (Tertile2) | 2017-2018 | USA | NAFLD patients and controls | case-control study | Transient elastography | | 2057/2132 | 67 | 30 | NA | 4189 | NAFLD | 7 |
| Li2022a (Quartile2) | 2011-2018 | USA | NAFLD patients and controls | case-control study | hepatic steatosis index | 24-h dietary recalls | 6259/6151 | 49 | 29.4 | NA | 12410 | NAFLD | 7 |
| Li2022b (Quartile3) | 2011-2018 | USA | NAFLD patients and controls | case-control study | hepatic steatosis index | | 6259/6151 | 49 | 29.4 | NA | 12410 | NAFLD | 7 |
| Li2022c (Quartile4) | 2011-2018 | USA | NAFLD patients and controls | case-control study | hepatic steatosis index | | 6259/6151 | 49 | 29.4 | NA | 12410 | NAFLD | 7 |
| Heidari2022 | 2019-2020 | NR | NAFLD patients and controls | case-control study | abdominal ultrasound | food frequency questionnaire | 140/100 | 37 | 25.9 | NA | 240 | NAFLD | 7 |
| Tian2022a (Tertile2) | 2017-2018 | USA | NAFLD patients and controls | case-control study | transient elastography | 24-h dietary recalls | NA | 48 | 31 | NA | 3573 | NAFLD | 7 |
| Tian2022b (Tertile3) | 2017-2018 | USA | NAFLD patients and controls | case-control study | transient elastography | | NA | 48 | 31 | NA | 3573 | NAFLD | 7 |
| Li2023a (Quartile2) | 2011-2018 | USA | NAFLD patients and controls | case-control study | fatty liver index | 24-h dietary recalls | 2765/2741 | 48 | 29.5 | NA | 5506 | Fibrosis | 7 |
| Li2023b (Quartile3) | 2011-2018 | USA | NAFLD patients and controls | case-control study | fatty liver index | | 2765/2741 | 48 | 29.5 | NA | 5506 | Fibrosis | 7 |
| Li2023c (Quartile4) | 2011-2018 | USA | NAFLD patients and controls | case-control study | fatty liver index | | 2765/2741 | 48 | 29.5 | NA | 5506 | Fibrosis | 7 |
| Zhang2023a (Quartile2) | 2005-2016 | USA | NAFLD patients and controls | case-control study | Fatty liver index NAFLD fibrosis score | energy-standardized version of world | 5070/4982 | 46 | 29 | NA | 10052 | NAFLD Fibrosis | 7 |
| Zhang2023b (Quartile3) | 2005-2016 | USA | NAFLD patients and controls | case-control study | Fatty liver index NAFLD fibrosis score | | 5070/4982 | 46 | 29 | NA | 10052 | NAFLD Fibrosis | 7 |
| Zhang2023c (Quartile4) | 2005-2016 | USA | NAFLD patients and controls | case-control study | Fatty liver index NAFLD fibrosis score | | 5070/4982 | 46 | 29 | NA | 10052 | NAFLD Fibrosis | 7 |
| Shi2023a (Tertile2) | 2005-2016 | USA | NAFLD patients and controls | case-control study | fatty liver index/He-patic Steatosis Index | 24-h dietary recalls food frequency questionnaire | 843/743 | 69 | 33.11 | NA | 1586 | Frailty | 8 |
| Shi2023b (Tertile3) | 2005-2016 | USA | NAFLD patients and controls | case-control study | fatty liver index/He-patic Steatosis Index | questionnaire | 843/743 | 69 | 33.11 | NA | 1586 | Frailty | 8 |

*(Continued)*

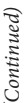

**Table 1.** (Continued)

| Author | Study period | Country | Population | Study design | Diagnostic criteria | DII measurement method | Male/Female | Mean age (yrs) | Mean BMI | DII cut-off | Patients NO | Outcome Measure | Quanlity score |
|---|---|---|---|---|---|---|---|---|---|---|---|---|---|
| Petermann-Rocha2023a (Very/moderately) | 2006-2010 | UK | general UK population | cohort study | NA | Oxford WebQ | 77581/93963 | 56 | NA | NA | 171544 | NAFLD | 7 |
| Petermann-Rocha2023b (Trend) | 2006-2010 | UK | general UK population | cohort study | NA | | 77581/93963 | 56 | NA | NA | 171544 | NAFLD | 7 |
| Valibeygi2023 | 2014-2019 | Iran | inhabitants | cohort study | fatty liver index | food frequency questionnaire | 4299/5493 | 57 | 29 | NA | 9792 | NAFLD | 7 |
| Soltanieh2023a | NA | Iran | type two diabetes | case-control study | transient elastography | food frequency questionnaire | 92/108 | 52 | 28 | -3.11 | 200 | NAFLD | 8 |
| Soltanieh2023b | NA | Iran | type two diabetes | case-control study | transient elastography | | 92/108 | 52 | 28 | -2.49 | 200 | NAFLD | 8 |
| Li2024a (Quartile2) | 2003-2018 | USA | NAFLD patients and controls | case-control study | Fatty liver index NAFLD fibrosis score | 24-h dietary recalls | 2563/2589 | 53 | 28 | NA | 5152 | NAFLD Fibrosis | 7 |
| Li2024b (Quartile3) | 2003-2018 | USA | NAFLD patients and controls | case-control study | Fatty liver index NAFLD fibrosis score | | 2563/2589 | 53 | 28 | NA | 5152 | NAFLD Fibrosis | 7 |
| Li2024c (Quartile4) | 2003-2018 | USA | NAFLD patients and controls | case-control study | Fatty liver index NAFLD fibrosis score | | 2563/2589 | 53 | 28 | NA | 5152 | NAFLD Fibrosis | 7 |
| Xu2024a | 2017–2020 | USA | NAFLD patients and controls | case-control study | transient elastography | 24-h dietary recalls | 2195/2495 | 49 | 29.89 | 0.09 | 4690 | NAFLD Fibrosis | 7 |
| Xu2024b | 2017–2020 | USA | NAFLD patients and controls | case-control study | transient elastography | | 2195/2495 | 49 | 29.89 | 1.35 | 4690 | NAFLD Fibrosis | 7 |
| Xu2024c | 2017–2020 | USA | NAFLD patients and controls | case-control study | transient elastography | | 2195/2495 | 49 | 29.89 | 2.61 | 4690 | NAFLD Fibrosis | 7 |
| Yan2024a | 2017–2018 | USA | NAFLD patients and controls | case-control study | transient elastography | 24-h dietary recalls | 1777/1856 | 49 | 31 | -0.26 | 3633 | NAFLD | 7 |
| Yan2024b | 2017–2018 | USA | NAFLD patients and controls | case-control study | transient elastography | | 1777/1856 | 49 | 31 | 1.08 | 3633 | NAFLD | 7 |
| Yan2024c | 2017–2018 | USA | NAFLD patients and controls | case-control study | transient elastography | | 1777/1856 | 49 | 31 | 2.28 | 3633 | NAFLD | 7 |
| Motamedi2024a (Quartile2) | 2016-2019 | Iran | patients referred to a university-affiliated nutrition | case-control study | fatty liver index | food frequency questionnaire | 1476/1682 | 40 | 41 | NA | 3158 | NAFLD | 7 |
| Motamedi2024b (Quartile3) | 2016-2019 | Iran | patients referred to a university-affiliated nutrition | case-control study | fatty liver index | | 1476/1682 | 40 | 41 | NA | 3158 | NAFLD | 7 |
| Motamedi2024c (Quartile4) | 2016-2019 | Iran | patients referred to a university-affiliated nutrition | case-control study | fatty liver index | | 1476/1682 | 40 | 41 | NA | 3158 | NAFLD | 7 |
| Hu2024a (Quartile2) | 2017-2018 | USA | NAFLD patients and controls | case-control study | transient elastography | 24-h dietary recalls | 1092/899 | 44 | 29 | NA | 1991 | NAFLD | 7 |
| Hu2024b (Quartile3) | 2017-2018 | USA | NAFLD patients and controls | case-control study | transient elastography | | 1092/899 | 44 | 29 | NA | 1991 | NAFLD | 7 |
| Hu2024c (Quartile4) | 2017-2018 | USA | NAFLD patients and controls | case-control study | transient elastography | | 1092/899 | 44 | 29 | NA | 1991 | NAFLD | 7 |

*(Continued)*

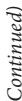

**Table 1.** (Continued)

| Author | Study period | Country | Population | Study design | Diagnostic criteria | DII measurement method | Male/ Female | Mean age (yrs) | Mean BMI | DII cut-off | Patients NO | Outcome Measure | Quanlity score |
|---|---|---|---|---|---|---|---|---|---|---|---|---|---|
| Doustmoham-madian2024a (Tertile2-Women) | 2016-2017 | Iran | 25 health centers in rural areas and 16 health centers in urban areas in the Amol City area, located in the northern region of Iran | case-control study | abdominal ultrasonography | food frequency questionnaire | 1452/1389 | 47 | 28 | NA | 3110 | NAFLD | 7 |
| Doustmoham-madian2024b (Tertile3-Women) | 2016-2017 | Iran | 25 health centers in rural areas and 16 health centers in urban areas in the Amol City area, located in the northern region of Iran | case-control study | abdominal ultrasonography | | 1452/1389 | 47 | 28 | NA | 3110 | NAFLD | 7 |
| Doustmoham-madian2024c (Tertile2-Men) | 2016-2017 | Iran | 25 health centers in rural areas and 16 health centers in urban areas in the Amol City area, located in the northern region of Iran | case-control study | abdominal ultrasonography | | 1452/1389 | 47 | 28 | NA | 3110 | NAFLD | 7 |
| Doustmoham-madian2024d (Tertile3-Men) | 2016-2017 | Iran | 25 health centers in rural areas and 16 health centers in urban areas in the Amol City area, located in the northern region of Iran | case-control study | abdominal ultrasonography | | 1452/1389 | 47 | 28 | NA | 3110 | NAFLD | 7 |

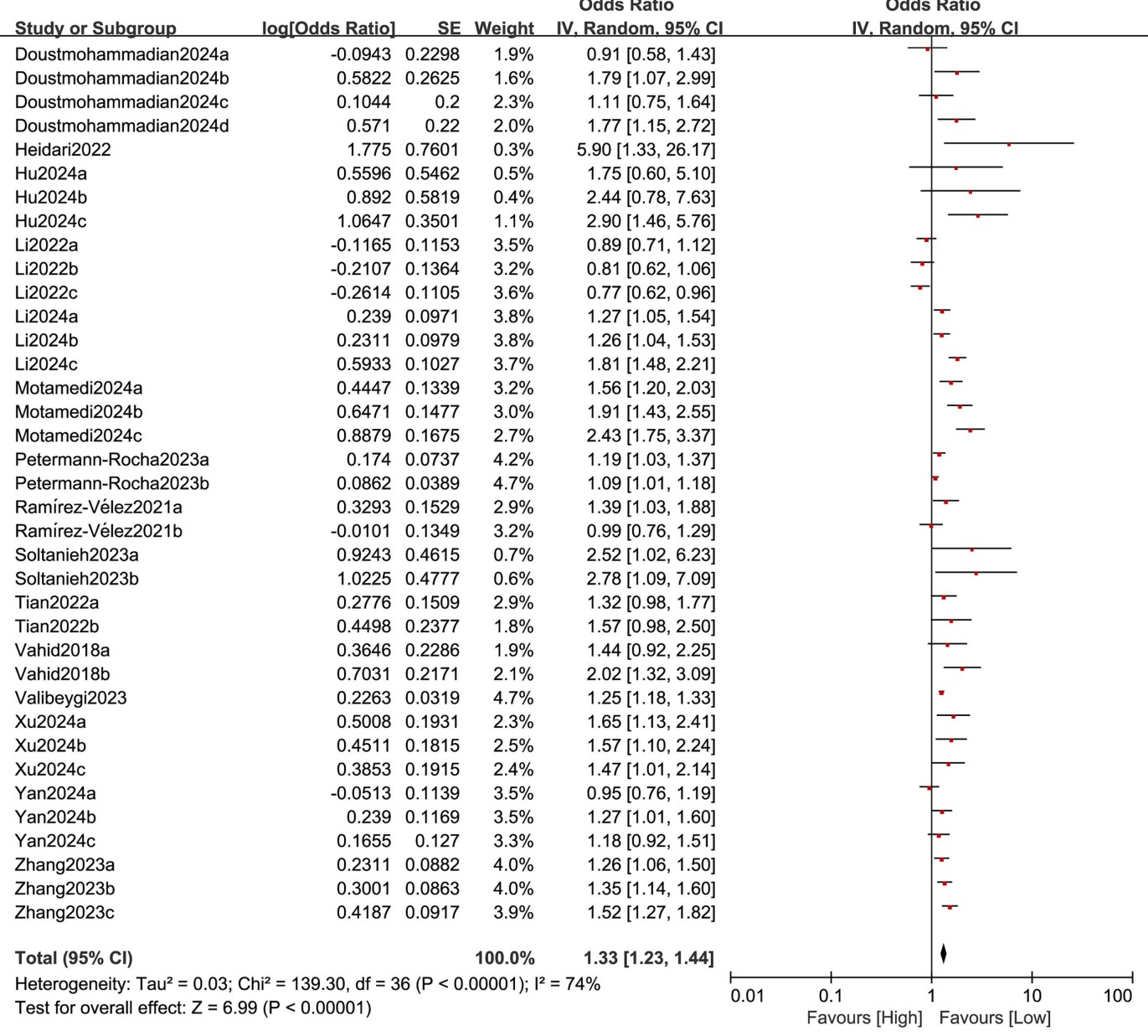

**Fig 2. Forest plot for the association between the DII and NAFLD.**

## 4. Discussion

Our meta-analysis of 18 studies with 262,468 adults and 37 comparison groups found an important relationship between high DII scores and NAFLD. However, substantial heterogeneity was observed (I²=74% in the NAFLD analysis), which may be attributable to several factors. We believe that this heterogeneity mainly arises from methodological and population-level factors. First, differences in DII calculation methods between studies may be a major influencing factor. The included studies used various dietary assessment tools, including food frequency questionnaires (FFQs) and 24-hour

**Table 2. Subgroup analysis of NAFLD and Fibrosis according to average age, study design, sample size, geographic region, and diagnostic criteria.**

| Subgroup | NAFLD | | | | Fibrosis | | | |
|---|---|---|---|---|---|---|---|---|
| | Study | OR [95%CI] | Z-test P value | I² | Study | OR [95%CI] | Z-test P value | I² |
| **Total** | 37 | 1.33 [1.23,1.44] | < 0.00001 | 74% | 12 | 1.36 [1.20-1.54] | < 0.00001 | 0% |
| **Study design** | | | | | | | | |
| Case-control | 34 | 1.38 [1.24,1.53] | < 0.00001 | 73% | / | / | / | / |
| Cohort study | 3 | 1.18 [1.06,1.30] | 0.001 | 74% | / | / | / | / |
| **Region** | | | | | | | | |
| Asia | 12 | 1.61 [1.34,1.93] | < 0.00001 | 71% | / | / | / | / |
| Europe | 2 | 1.11 [1.03,1.20] | 0.004 | 10% | / | / | / | / |
| America | 22 | 1.26[1.13,1.41] | < 0.0001 | 73% | / | / | / | / |
| **Average age** | | | | | | | | |
| > 50 | 10 | 1.28 [1.15,1.41] | < 0.00001 | 73% | 3 | 1.39[1.13,1.71] | 0.002 | 0% |
| ≤50 | 27 | 1.37 [1.22,1.55] | < 0.00001 | 75% | 9 | 1.32[1.11,1.57] | 0.002 | 17% |
| **Sample size** | | | | | | | | |
| > 5000 | 12 | 1.19[1.07,1.31] | 0.001 | 83% | 9 | 1.40[1.23,1.60] | < 0.00001 | 0% |
| ≤5000 | 25 | 1.50[1.32,1.70] | < 0.00001 | 60% | 3 | 1.03[0.71,1.50] | 0.88 | 0% |
| **Diagnostic criteria** | | | | | | | | |
| Ultrasonography | 7 | 1.51[1.14,1.98] | 0.003 | 54% | 3 | 1.15[0.88,1.50] | 0.31 | 0% |
| Tatty liver index | 13 | 1.31[1.15,1.49] | < 0.0001 | 85% | 6 | 1.50[1.29,1.75] | < 0.00001 | 0% |
| Transient elastography | 15 | 1.37[1.19,1.57] | < 0.00001 | 49% | 3 | 1.03[0.71,1.50] | 0.88 | 0% |

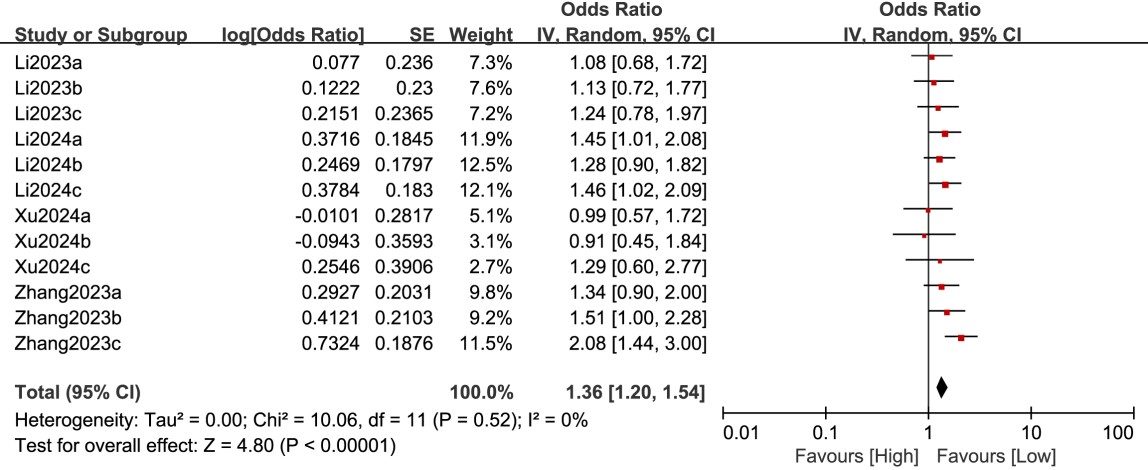

**Fig 3. Forest plot for the association between the DII and fibrosis.**

dietary recalls. These methods differ in their focus on daily intake habits and the estimation of nutritional parameters, which may lead to misjudgment of an individual's pro-inflammatory dietary potential. In addition, the number of food parameters used to calculate DII scores varied across studies. This inconsistency in input data directly affects the comparability and absolute values of DII scores between different cohorts, leading to measurement heterogeneity. Second, the diversity of baseline characteristics and the adjustment for confounding factors are key sources of heterogeneity. The

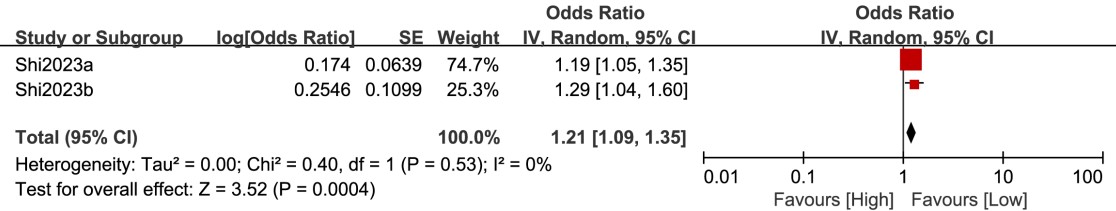

**Fig 4. Forest plot for the association between the DII and frailty.**

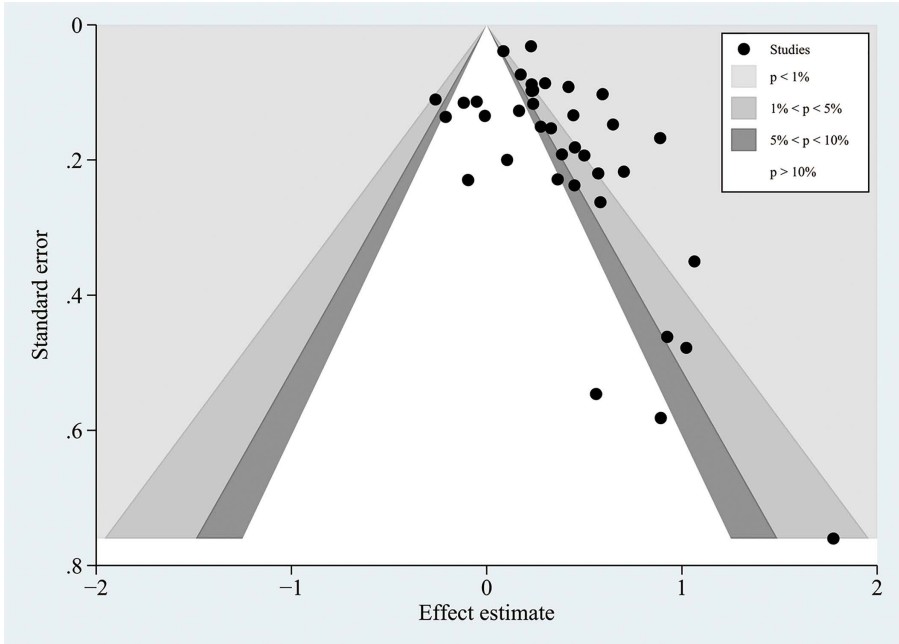

**Fig 5. Funnel plot for the evaluation of publication bias in NAFLD.**

populations included in the studies varied geographically (Asia, Europe, Americas) and had different dietary patterns, genetic backgrounds, and other characteristics. Although we adjusted for major confounders such as age and BMI, the degree and consistency of adjustment for other important factors (such as physical activity, detailed medication use, and socio-economic status) differed across studies. Residual confounding from these unmeasured or inadequately adjusted variables may amplify or attenuate the observed effect sizes to varying degrees. Subgroup analysis of regional and dietary patterns showed that Asians had a significantly higher odds ratio (OR = 1.61) than Europeans (OR = 1.11) and the Americans (OR = 1.26). This disparity likely results from multiple factors, including lifestyle, dietary habits, and genetic predisposition. In terms of dietary composition, Asians primarily consume refined carbohydrates, such as white rice, wheat flour, and rice noodles, whereas Europeans have a more diverse range of carbohydrate sources. Although red meat consumption is lower in Asians compared to Western populations, their preference for grilled or processed meats—commonly referred to as "char-grilled" foods—may increase the risk of liver inflammation. Moreover, Asians frequently use traditional oils such as soybean and palm oil, which are high in polyunsaturated fatty acids, while consuming less fish and olive oil. A higher intake of polyunsaturated fats has been reported to correlate positively with an increased risk of non-alcoholic

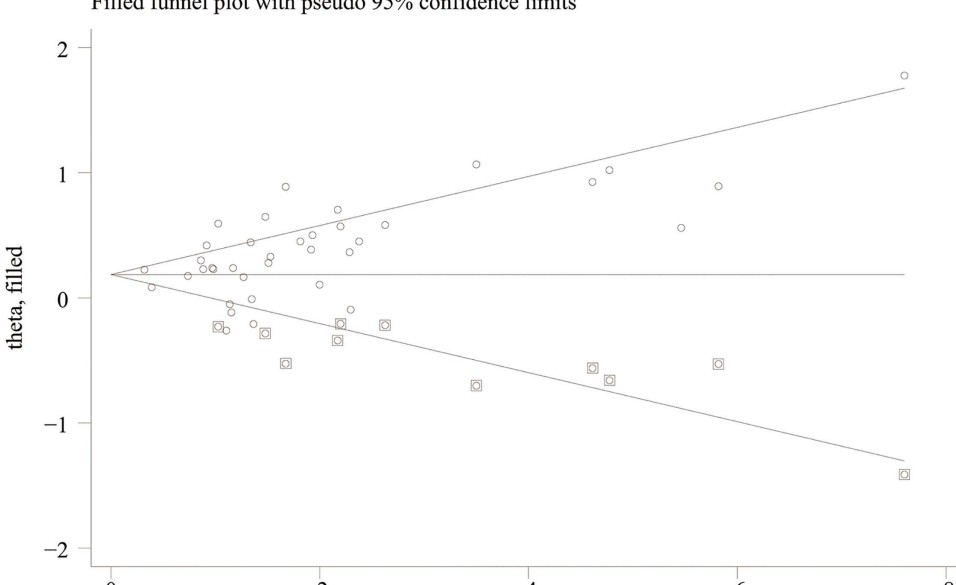

**Fig 6. Filled funnel plot for the evaluation of publication bias in NAFLD.**

fatty liver disease (NAFLD). Excessive caloric intake and high levels of saturated fats and carbohydrates contribute to the development of hepatic steatosis [46]. The lipolytic breakdown of triglycerides elevates free fatty acid levels, while excess carbohydrates—particularly fructose—are converted into fatty acids by liver cells. Overloaded fatty acids undergo β-oxidative metabolism or de novo lipogenesis, triggering endoplasmic reticulum stress, oxidative stress, and inflammasome activation [47]. The inclusion of different age groups (e.g., ≤ 50 vs. > 50 years) may amplify the effects of the DII, particularly due to metabolic disorders and chronic inflammation in older adults. Furthermore, varying adjustments for confounding factors, such as obesity and diabetes, may contribute to differences in effect sizes. Although all studies reported NOS scores ≥6, significant differences in study design and sample size were observed, with larger studies likely providing more stable effect estimates, whereas smaller studies may be more susceptible to residual confounding. The DII is based on its effect on inflammatory markers (e.g., CRP, IL-6), but inconsistent measurement methods and cutoff values across studies may affect its association with NAFLD. Sensitivity analyses confirmed the consistency of the results; however, funnel plot asymmetry and Egger's test indicated a potential bias, likely due to unpublished negative findings. These observations underscore the need for further research to validate the association between DII and NAFLD. Subgroup analyses identified geographic regions and diagnostic criteria as sources of heterogeneity, with European studies showing reduced heterogeneity (10%), suggesting greater reliability, possibly due to more homogeneous Western dietary patterns.

The relationship between the DII and liver fibrosis in adults was examined in 12 comparison groups. The funnel plot showed symmetry, and Egger's test indicated no bias, suggesting high-quality evidence. Subgroup analysis revealed that larger sample sizes strengthened the association between the DII and the progression of liver fibrosis, confirming a positive correlation between the DII and liver fibrosis risk in NAFLD patients. These results support the robust and consistent correlation between the DII and liver fibrosis.

Our meta-analysis shows significant associations between higher DII scores and NAFLD incidence (OR = 1.33) and fibrosis (OR = 1.36), though the clinical relevance of these findings requires careful consideration. A 30–40% increased risk, while modest at the individual level, may have a substantial population-level impact due to NAFLD's high prevalence.

This effect size is comparable to other dietary risk factors, such as excessive fructose intake (OR ~1.2–1.5) and low fiber consumption (OR ~1.3) in NAFLD development. Clinically, these results support incorporating DII-based assessment into NAFLD risk stratification, particularly for high-risk groups (e.g., patients with metabolic syndrome). A pro-inflammatory diet (DII > +1) could signal the need for earlier hepatic screening or lifestyle intervention. Public health strategies should focus on reducing inflammatory dietary components [48], and promoting anti-inflammatory foods in NAFLD prevention programs. However, further research is necessary to determine whether DII-targeted dietary modifications improve liver histology or induce fibrosis regression.

Although Zhang et al. [15] confirmed that a high DII is linked to an increased risk of NAFLD, our study not only supports their conclusion that the DII is positively associated with NAFLD but also suggests that pro-inflammatory diets may accelerate the progression of liver disease (e.g., fibrosis), with potential implications for clinical intervention. We included newer and larger studies, and we performed subgroup analyses. It was observed through subgroup analysis that the heterogeneity of NAFLD was primarily due to geographic region and diagnostic criteria. By incorporating more studies and analyzing additional indicators, we strengthened the association between the DII advanced liver fibrosis as well as non-alcoholic cirrhosis. Subgroup analyses further addressed heterogeneity, reinforcing our findings. Thus, our study provides stronger evidence than previous research.

The pathogenesis of NAFLD involves various factors, including diet, physical activity, and inflammation [49,50]. Dietary patterns have a significant impact on the development and regulation of NAFLD-related inflammation [51–53]. Several studies have shown that chronic inflammatory markers, such as serum C-reactive protein (CRP), procalcitonin, interleukin-6 (IL-6), and tumor necrosis factor-α (TNF-α), contribute to NAFLD [9]. Certain dietary compounds can influence inflammation [52], potentially affecting the progression of NAFLD. One limitation of focusing on individual foods or nutrients is that they are typically consumed in combination, and their interactions can influence their overall impact [29]. The DII considers both pro-inflammatory and anti-inflammatory foods. Previous studies have shown that the DII correlates with inflammatory markers, including CRP, IL-6, TNF-α, and homocysteine [54–56]. The DII evaluates a diet's inflammatory potential across different populations by analyzing individual intake through tools such as 24-hour recall, Food Frequency Questionnaires (FFQs), 7-day dietary recall (7DDR), and diet diaries [57]. Lower scores on the DII correspond to anti-inflammatory dietary patterns, whereas higher scores are associated with pro-inflammatory diets. [9]. Dietary patterns and inflammatory imbalances play a central role in the advancement of NAFLD [7,58–60]. An evidence-based synthesis through systematic review and meta-analysis was conducted to determine the impact of the DII on NAFLD incidence and prognosis, thereby providing evidence for its clinical utility in identifying high-risk patients with poor prognosis.

## 5. Limitations

Our meta-analysis has several limitations. First, moderate heterogeneity persisted despite using a random-effects model, which may affect the reliability of the results. This heterogeneity could result from inconsistent DII calculation methods, differences in baseline characteristics of study populations, and inadequate adjustment for confounding factors (e.g., lifestyle, comorbidities). Therefore, further validation in a more standardized, large cohort study is needed to clarify the role of the DII in NAFLD progression. Second, publication bias is a concern and may arise from small-sample studies with negative results not being published. However, after adjusting using the trim-and-fill method, the effect size remained significant (OR=1.205), indicating that the results of DII and NAFLD remain stable, suggesting a possible association between the two, future research needs to correct for methodological differences to confirm the strength of the association. Additionally, the limited research on the DII's effect on frailty incidence prevented its inclusion in our analysis, and future large-sample prospective studies are needed. Third, our analysis shares the geographic limitations common to many systematic reviews in this field, with a predominance of studies from regions such as North America and Western Europe. This clustering may affect the generalizability of our findings due to factors, such as regional nutritional habits, food availability, population-specific genetic factors, and differences in healthcare access, screening practices, and treatment protocols. While we

attempted to account for these factors through subgroup analyses, the limited representation from regions like South Asia, Africa, and South America means that our findings should be interpreted cautiously in these populations. Future research should prioritize geographic diversity to validate our findings across different settings. Fourth, our analysis could not assess potential dose-response relationships between DII scores and outcomes. Although such an analysis would strengthen evidence for causality, the available data were insufficient. Most studies reported DII in categorical formats (e.g., tertiles or quartiles), and few provided detailed data on incremental changes in DII. Future research should explicitly evaluate dose–response patterns and assess whether DII-guided dietary interventions improve NAFLD prognosis, thereby enhancing our understanding of the relationship between higher DII scores and increased health risks. Fifth, limitations exist in the definition of cut-off values for the DII. Due to inconsistent classification criteria across studies (e.g., quartiles, medians, or specific thresholds), we could not harmonize the clinical definition of a high DII. This inconsistency may affect cross-sectional comparisons and limit the DII's clinical applicability in NAFLD prognostication. Future studies should establish standardized DII cut-off values through large-scale population data to guide individualized dietary interventions. Sixth, many studies were not fully accessible, primarily due to limited access to proprietary databases, incomplete records, or publication in non-indexed journals, which may introduce selection bias. Seventh, although the 2023 consensus recommends using the term MASLD, since our analysis is based on historical studies using NAFLD criteria, this paper continues to use the term NAFLD to maintain consistency with the original studies; this terminology difference does not affect the analysis of fatty liver pathology itself. Future research adopting the MASLD framework may refine diagnostic precision and patient stratification, potentially affecting the reproducibility of our results. Nevertheless, our conclusions remain valid within the context of NAFLD-defined populations, which remain clinically relevant in hepatology practice.

## 6. Conclusion

In summary, our study found that a pro-inflammatory diet (high DII) may be associated with an increased incidence of NAFLD and a higher risk of progression to fibrosis. These findings highlight the potential utility of the DII as a practical tool for identifying high-risk dietary patterns in clinical practice. However, there are several limitations in this study, including moderate heterogeneity, publication bias, insufficient geographical representation (mainly in Europe and the United States), the lack of a dose-response relationship, and inconsistent cut-off values for the DII. These issues may affect the reliability and generalizability of the results and should be addressed in more standardized and geographically diverse studies in the future.

## Supporting information

**S1 Table. PRISMA checklist.**
(DOCX)

**S2 Table. Literature search strategy.**
(DOCX)

**S3 Table. Quality evaluation of the eligible studies with Newcastle–Ottawa scale.**
(DOCX)

**S1 Fig. Sensitivity analysis of DII and NAFLD.**
(TIF)

**S2 Fig. Sensitivity analysis of DII and Fibrosis.**
(TIF)

**S3 Fig. Funnel plot for the evaluation of publication bias for Fibrosis.**
(TIF)

## Author contributions

**Conceptualization:** Jingjing Lin.

**Data curation:** Jingjing Lin.

**Formal analysis:** Jingjing Lin, Lina Shen.

**Investigation:** Lina Shen.

**Methodology:** Jingjing Lin.

**Project administration:** Jingjing Lin.

**Resources:** Jingjing Lin, Lina Shen.

**Software:** Mengna Huang, Lina Shen.

**Supervision:** Mengna Huang.

**Validation:** Mengna Huang.

**Visualization:** Mengna Huang.

**Writing – original draft:** Jingjing Lin, Mengna Huang, Lina Shen.

**Writing – review & editing:** Jingjing Lin.

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
