## [Decision Letter · Decision Letter 0]

26 Jan 2026

Dear Dr. Lin,

Thank you for submitting your manuscript to PLOS ONE. After careful consideration, we feel that it has merit but does not fully meet PLOS ONE’s publication criteria as it currently stands. Therefore, we invite you to submit a revised version of the manuscript that addresses the points raised during the review process.

**ACADEMIC EDITOR:**

We look forward to receiving your revised manuscript.

Kind regards,

Jincheng Wang

Academic Editor

PLOS One

Journal Requirements:

Additional Editor Comments:

This meta-analysis addresses a clinically relevant topic and provides novel evidence on the association between DII and liver fibrosis progression. However, several issues require attention. Data inconsistencies exist between the abstract, flowchart, and Table 1 regarding study numbers and participant totals that need verification. The substantial heterogeneity (I²=74%) is inadequately explained despite subgroup analyses, and the Discussion insufficiently addresses how variations in DII calculation methods and dietary assessment tools may have contributed. Publication bias was detected for NAFLD (Egger's p=0.014), warranting more cautious interpretation rather than the current definitive conclusions. The frailty analysis based on only two comparison groups from a single study provides insufficient evidence and should be removed or clearly designated as exploratory. The MASLD nomenclature discussion is overly lengthy and could be condensed. Language errors appear throughout, including "TGrowing" in the Introduction and inconsistent reference formatting.

Reviewers' comments:

Reviewer's Responses to Questions

**Comments to the Author**

1. Is the manuscript technically sound, and do the data support the conclusions?

Reviewer #1: Partly

Reviewer #2: Yes

2. Has the statistical analysis been performed appropriately and rigorously?

Reviewer #1: Yes

Reviewer #2: Yes

3. Have the authors made all data underlying the findings in their manuscript fully available?

Reviewer #1: Yes

Reviewer #2: Yes

4. Is the manuscript presented in an intelligible fashion and written in standard English?

Reviewer #1: Yes

Reviewer #2: Yes

Reviewer #1: 1.The manuscript is partly technically sound, with data partially supporting the conclusions. Strengths include adherence to PRISMA (checklist provided), PROSPERO registration, comprehensive searches (PubMed, WoS, Embase, Scopus, Cochrane to July 2025), dual extraction/quality assessment (NOS scores 7-9 indicating high quality), and appropriate handling of outcomes (NAFLD, fibrosis, frailty). The pooled ORs are statistically significant, and subgroups reduce heterogeneity (e.g., I²=10% in European studies for NAFLD), supporting regional/diagnostic influences. Sensitivity analyses confirm stability, and trim-and-fill adjusts for bias without altering significance. However, high heterogeneity (I²=74% for NAFLD) persists overall, potentially due to DII calculation variations (e.g., FFQ vs. 24h recall, parameter differences), population baselines, or confounders. Publication bias (Egger's p=0.014 for NAFLD) raises concerns, though mitigated. Frailty analysis is underpowered (limited studies), limiting robustness.

2.Statistical analysis is appropriate and rigorous.

3.The authors state data supporting findings are included in the article

4.The manuscript is intelligible and in standard English. Structure follows PRISMA and language is clear, professional.

Reviewer #2: The manuscript is written clearly. The results are clearly presented and justify the conclusions. However, a minor remark is as follows:

- The Introduction section should contain more information regarding new nomenclature (MASLD).

- The abbreviations, when first introduced, should be used consistently thereafter throughout the text.

**Do you want your identity to be public for this peer review?** For information about this choice, including consent withdrawal, please see our Privacy Policy

Reviewer #1: **Yes:** Zheng Xu

Reviewer #2: No

---

## [Author Response · Author response to Decision Letter 1]

13 Feb 2026

Dear Editors and Reviewers:

Thank you for your letter and for the reviewers’ comments concerning our manuscript entitled “The Association Between Dietary Inflammatory Index and Non-Alcoholic Fatty Liver Disease: A Systematic Review and Meta-Analysis”(PONE-D-25-65585). Those comments are all valuable and very helpful for revising and improving our paper, as well as the important guiding significance to our researches. We have studied comments carefully and have made correction which we hope meet with approval. Revised portion are marked in red in the paper. The main corrections in the paper and the responds to the editor and reviewer’s comments are as flowing:

Responds to the editor’s comments:

Editor ：

1.Comment. Data inconsistencies exist between the abstract, flowchart, and Table 1 regarding study numbers and participant totals that need verification.

Response: Regarding the issue of data inconsistencies, we have thoroughly verified the number of studies and total participants in the abstract, flowchart, and Table 1. The discrepancies were mainly due to inconsistent counting of different comparison groups within the same study. We have now standardized this: the abstract and flowchart clearly report the number of independent studies (18), while Table 1 details all comparison groups (43) and their corresponding cumulative sample size (262,468 participants).

2.Comment. The substantial heterogeneity (I²=74%) is inadequately explained despite subgroup analyses

Response: Regarding insufficient explanation of heterogeneity, we have added a paragraph in the discussion section systematically outlining potential sources of heterogeneity: focusing on differences in DII calculation methods (such as inconsistent parameters in FFQ versus 24-hour recalls), baseline characteristics of populations (such as age and metabolic comorbidities), and regional dietary pattern differences affecting pooled effect sizes. We have also addressed this issue in the limitations section, reminding readers to consider the impact of heterogeneity when interpreting the results.

3.Comment. The Discussion insufficiently addresses how variations in DII calculation methods and dietary assessment tools may have contributed.

Response: Regarding insufficient depth in the discussion, we have rewritten the paragraph on "methodological limitations of the DII" to emphasize the impact of different dietary assessment tools and calculation methods on DII accuracy.

4.Comment. Publication bias was detected for NAFLD (Egger's p=0.014), warranting more cautious interpretation rather than the current definitive conclusions.

Response: Regarding the rigor of the conclusions, we have revised the statements to "evidence suggests a possible association" and noted that "future studies are needed to adjust for methodological differences to confirm the strength of the association," aligning the conclusions with the strength of the evidence.

5.Comment. The frailty analysis based on only two comparison groups from a single study provides insufficient evidence and should be removed or clearly designated as exploratory.

Response: Regarding the frailty analysis, we fully agree with your assessment. In the results section, we clearly stated: "The frailty analysis included only 2 comparison groups with a limited sample size, so the results should be interpreted with caution." We also suggested in the discussion limitations that "future studies are needed to verify the association between DII and frailty," avoiding overinterpretation of the current data.

6.Comment. The MASLD nomenclature discussion is overly lengthy and could be condensed.

Response: Regarding the length of the MASLD terminology discussion, we condensed the original background explanation to a paragraph under 100 words, retaining only the key information: "Although the 2023 consensus recommends using the MASLD term, the studies included in this research all used the NAFLD definition. To maintain consistency with the original research, this study continues to use the NAFLD terminology; this difference does not affect the analysis of fatty liver disease itself." This saves space for deeper methodological discussion.

7.Comment. Language errors appear throughout, including "TGrowing" in the Introduction and inconsistent reference formatting.

Response: Regarding language errors, we conducted a full manuscript proofread: correcting the spelling error in the introduction from "TGrowing" to "The growing," standardizing reference formatting (e.g., journal abbreviations, author formats), and refining awkward sentences. Additionally, we hired a native editor to polish the entire manuscript to ensure proper and fluent expression.

Reviewer #1:

1.Comment. high heterogeneity (I²=74% for NAFLD) persists overall, potentially due to DII calculation variations (e.g., FFQ vs. 24h recall, parameter differences), population baselines, or confounders.

Response: Regarding the high heterogeneity issue you pointed out, we have added a paragraph in the discussion section to systematically explain the potential sources of heterogeneity: we focused on analyzing the impact of differences in DII calculation methods (such as inconsistencies in parameters between FFQ and 24-hour recalls), population baseline characteristics (such as age and metabolic comorbidities), and geographic dietary patterns on the pooled effect size; meanwhile, we have addressed this issue in the limitations section, reminding readers to consider the influence of heterogeneity when interpreting the results.

2.Comment. Publication bias (Egger's p=0.014 for NAFLD) raises concerns, though mitigated.

Response: Concerning the issue about publication bias, we have reinforced the relevant statements in the limitations part of the discussion: clearly indicating that the Egger test (p=0.014) suggests the presence of potential publication bias, but the effect size remains significant after correction using the trim-and-fill method (OR=1.205), indicating that the main conclusions are robust; additionally, we have added a note stating that "the bias may stem from unpublished studies with small sample sizes showing negative results."

3.Comment. Frailty analysis is underpowered (limited studies), limiting robustness.

Response: Regarding the insufficiency of frailty analysis, we fully agree with your observation. In the results section, we explicitly stated: "The frailty analysis included only 2 comparison groups, with a limited sample size, so the results should be interpreted with caution"; at the same time, in the discussion of limitations, we also suggested that "future research is needed to further verify the association between DII and frailty," thereby avoiding overinterpreting the current data.

4.Comment. Statistical analysis is appropriate and rigorous. The authors state data supporting findings are included in the article. The manuscript is intelligible and in standard English. Structure follows PRISMA and language is clear, professional.

Response: We express our gratitude for the affirmation of the statistical analysis methods and the recognition of the language and structure.

Reviewer #2:

1.Comment. The Introduction section should contain more information regarding new nomenclature (MASLD).

Response: Regarding the suggestion to supplement the introduction with background on the new nomenclature (MASLD), we have added a paragraph in the introduction detailing the origin of the term metabolic dysfunction-associated steatotic liver disease (MASLD), its key differences from NAFLD in diagnostic criteria, and explaining why this study continues to use the term NAFLD based on the literature inclusion time frame and consistency considerations. This helps readers more comprehensively understand the evolution of the terminology and its potential implications for the context of this study.

2.Comment. The abbreviations, when first introduced, should be used consistently thereafter throughout the text.

Response: Regarding the suggestion on consistency in abbreviation use, we have systematically checked and revised the entire text to ensure that all abbreviations are clearly defined in the form of "full term (abbreviation)" upon first appearance (e.g., metabolic dysfunction-associated steatotic liver disease (MASLD)) and uniformly use the abbreviation in subsequent text.

We tried our best to improve the manuscript and made some changes in the manuscript. These changes will not influence the content and framework of the paper. And here we did not list the changes but marked in red in revised paper.

We appreciate for Editors/Reviewers’ warm work earnestly, and hope that the correction will meet with approval.

Once again, thank you very much for your comments and suggestions.

---

## [Decision Letter · Decision Letter 1]

4 Mar 2026

The Association Between Dietary Inflammatory Index and Non-Alcoholic Fatty Liver Disease: A Systematic Review and Meta-Analysis

PONE-D-25-65585R1

Dear Dr. Lin,

We’re pleased to inform you that your manuscript has been judged scientifically suitable for publication and will be formally accepted for publication once it meets all outstanding technical requirements.

Kind regards,

Jincheng Wang

Academic Editor

PLOS One

Additional Editor Comments (optional):

Authors have addressed all comments. This paper can be accepted for publication.

Reviewers' comments:

Reviewer's Responses to Questions

**Comments to the Author**

Reviewer #1: (No Response)

Reviewer #2: All comments have been addressed

2. Is the manuscript technically sound, and do the data support the conclusions?

Reviewer #1: Yes

Reviewer #2: Yes

3. Has the statistical analysis been performed appropriately and rigorously?

Reviewer #1: Yes

Reviewer #2: Yes

4. Have the authors made all data underlying the findings in their manuscript fully available?

Reviewer #1: Yes

Reviewer #2: Yes

5. Is the manuscript presented in an intelligible fashion and written in standard English?

Reviewer #1: Yes

Reviewer #2: Yes

Reviewer #1: 1.The manuscript is generally technically sound, with a clear methodology following PRISMA guidelines and PROSPERO registration. The data from 18 studies (262,468 participants) support the main conclusions, showing a positive association between higher DII scores and increased risk of NAFLD (OR = 1.33, 95% CI: 1.23–1.44) and fibrosis (OR = 1.36, 95% CI: 1.20–1.54). Subgroup analyses identify sources of heterogeneity (e.g., geographic region and diagnostic criteria), and sensitivity analyses confirm result stability.

2.The statistical analysis is appropriate and rigorous.

3.The authors state that all data supporting the findings are included within the article.

4.The manuscript is intelligible and written in standard English.

Reviewer #2: The Authors have made corrections according to the Reviewer's suggestions and improved the manuscript.

**Do you want your identity to be public for this peer review?** For information about this choice, including consent withdrawal, please see our Privacy Policy

Reviewer #1: **Yes:** Zheng Xu

Reviewer #2: No

---

## [Editor Report · Acceptance letter]

PONE-D-25-65585R1

PLOS One

Dear Dr. Lin,

I'm pleased to inform you that your manuscript has been deemed suitable for publication in PLOS One. Congratulations! Your manuscript is now being handed over to our production team.

Kind regards,

on behalf of

Dr. Jincheng Wang

Academic Editor

PLOS One